# Development of Deep Learning Models for Real-Time Thoracic Ultrasound Image Interpretation

**DOI:** 10.3390/jimaging11070222

**Published:** 2025-07-05

**Authors:** Austin J. Ruiz, Sofia I. Hernández Torres, Eric J. Snider

**Affiliations:** Organ Support and Automation Technologies Group, U.S. Army Institute of Surgical Research, JBSA Fort Sam Houston, San Antonio, TX 78234, USAsofia.i.hernandeztorres.ctr@health.mil (S.I.H.T.)

**Keywords:** ultrasound imaging, artificial intelligence, deep learning, thoracic, emergency medicine, trauma

## Abstract

Thoracic injuries account for a high percentage of combat casualty mortalities, with 80% of preventable deaths resulting from abdominal or thoracic hemorrhage. An effective method for detecting and triaging thoracic injuries is point-of-care ultrasound (POCUS), as it is a cheap and portable noninvasive imaging method. POCUS image interpretation of pneumothorax (PTX) or hemothorax (HTX) injuries requires a skilled radiologist, which will likely not be available in austere situations where injury detection and triage are most critical. With the recent growth in artificial intelligence (AI) for healthcare, the hypothesis for this study is that deep learning (DL) models for classifying images as showing HTX or PTX injury, or being negative for injury can be developed for lowering the skill threshold for POCUS diagnostics on the future battlefield. Three-class deep learning classification AI models were developed using a motion-mode ultrasound dataset captured in animal study experiments from more than 25 swine subjects. Cluster analysis was used to define the “population” based on brightness, contrast, and kurtosis properties. A MobileNetV3 DL model architecture was tuned across a variety of hyperparameters, with the results ultimately being evaluated using images captured in real-time. Different hyperparameter configurations were blind-tested, resulting in models trained on filtered data having a real-time accuracy from 89 to 96%, as opposed to 78–95% when trained without filtering and optimization. The best model achieved a blind accuracy of 85% when inferencing on data collected in real-time, surpassing previous YOLOv8 models by 17%. AI models can be developed that are suitable for high performance in real-time for thoracic injury determination and are suitable for potentially addressing challenges with responding to emergency casualty situations and reducing the skill threshold for using and interpreting POCUS.

## 1. Introduction

Point-of-care ultrasound (POCUS) is a cost-effective, noninvasive diagnostic procedure that provides real-time medical imaging insights, packaged in portable ultrasound (US) systems that can be easily transported for field use. For the past two decades, the accessibility of POCUS systems has increased dramatically, resulting in a significant effect on the integration of POCUS across a variety of medical specialties [1]. Some of these specialties include critical care and emergency medicine, as the quick real-time insight display modality and the portability of POCUS systems in pre-hospital settings makes them invaluable for making rapid medical decisions required to triage effectively.

Despite the potential advantages of POCUS diagnostics, implementation is only possible with skilled radiologists, who may not be readily available in emergency situations to interpret injuries from US scans, such as during mass casualty situations that can occur far away from definitive care. This expected shortage of expertise can directly affect triage efforts, especially with time-sensitive potentially fatal injuries such as combat trauma in the thoracic region and abdominal region [2,3,4].

Artificial intelligence (AI) in medical imaging has gained rapid development in US image analysis by automating diagnosis tasks involving segmentation, object detection, quality assessment, and injury classification [5,6]. While these techniques can automate data interpretation in trauma cases, large datasets are needed to properly create algorithms for POCUS use cases. In response, a large swine US image dataset was captured for the thoracic and abdominal cavities with varying injury states for training deep learning (DL) models for DL classification, given their success in similar applications of identifying lung nodules in computed tomography scans, as well as the performance of classification architecture without the need for segmentation [7]. From this dataset, multi-class and binary classification models were developed for scan sites using custom-developed convolutional neural networks (CNNs) and preexisting CNN classification architectures. A recently conducted animal study explored the real-time performance of three-class thoracic motion mode (M-mode) models to distinguish between negative for injury (or baseline), pneumothorax (PTX), and hemothorax (HTX); example M-mode images for each class are highlighted in Figure 1. However, performance was significantly reduced during real-time implementation, with accuracies only reaching 67% for this thoracic application [8].

The objective of this research effort is to improve the real-time performance of DL models for thoracic detection of PTX and HTX injuries. Specifically, the goal was to demonstrate the utility of thoracic M-mode diagnostic AI suitability in real-time, ultimately for pairing with POCUS equipment in a far-forward environment. The contributions of this study are as follows:Additional swine data curation was implemented for a more robust dataset compared to previous studies.The filtering methods of the dataset were designed from the results of image preprocessing and analysis to correct data distribution shifts.A multi-class DL CNN was developed and optimized to classify ultrasound images as uninjured, HTX, or PTX.The model performance was evaluated with real-time data captures to highlight how the advancements improved performance when compared to previous studies.

## 2. Related Work

CNNs for classifying images or identifying objects of interest have become a common choice for medical image analysis, for their demonstrated success in the field [9]. For thoracic applications, Regions with CNN features (R-CNN) object detection models have been developed for identifying lung nodules on CT scans after lung nodules were labeled by subject matter experts for training the R-CNN models [10]. Similar approaches have been developed using only classification CNN models without the need for hand-curated segmentation of lung nodules, relying on automatic feature extraction from classification models to identify the appropriate regions of interest [11]. These kinds of DL models have only recently been applied to emergency medicine and POCUS applications [12,13,14], such as for the detection of free fluid in the intraperitoneal space in the right upper quadrant of the abdomen and pericardial space [15].

However, fewer studies have evaluated the use of AI for traumatic thoracic injury identification, with the exception of COVID-19 applications [16,17]. Kim et al. developed an EffecientNet-Lite0 model for assessing lung sliding in M-mode images and achieved approximately 95% area under the receiver operating characteristic curve scores [18]. However, this approach required stepwise interpretation of B-mode images prior to M-mode analysis. Montgomery et al. compared three different neural network architectures for detecting PTX in B-mode images at 86% sensitivity [19]. However, the image set was limited in size to only a single video per patient. Hannan et al. developed a YOLOv4-based model capable of working in real-time on a mobile application, but like the other related work, the application was not suitable for distinguishing HTX or pleural effusion from pneumothorax [20]. AI models for pleural effusion have been focused on COVID-19-related detection, as opposed to HTX originating from trauma.

We previously developed AI models for binary identification of PTX or HTX using B-mode or M-mode datasets captured in canines [21] or swine [7] using ShrapML CNN [22], MobileNetV2 [23], DarkNet53 [24], and Bayesian-optimized CNN architectures. Overall, MobileNetV2 performed most consistently, with an accuracy of approximately 90%, for M-mode images. We recently developed a three-class model to distinguish injury-negative, PTX, and HTX images in M-mode data using a YOLOv8 image classification model architecture [8]. This model was selected due to its synergy with object detection YOLO-based models being used for automated rib detection for streamlining data acquisitions, but performance in real time with blind datasets was low, at approximately 67%. As such, there is a need to improve three-class M-mode POCUS image interpretation with suitability for real-time use cases.

## 3. Materials and Methods

This section is organized into two main parts––readying the image dataset and DL model training and evaluation––following the block diagram depicted in Figure 2. The dataset referred to in this study is comprised of thoracic ultrasound (US) scans from previously captured swine US datasets [7,8]. Research was conducted in compliance with the Animal Welfare Act, implementing Animal Welfare regulations, and with the principles of the Guide for the Care and Use for Laboratory Animals. The Institutional Animal Care and Use Committee at the United States Army Institute of Surgical Research approved all research conducted in this study. The facility where this research was conducted is fully accredited by AAALAC International. Live animal subjects were maintained under a surgical plane of anesthesia and analgesia throughout the studies. Data were mainly captured at two time points: after subject instrumentation and post-euthanasia.

### 3.1. Readying the Image Dataset

#### 3.1.1. Dataset Preparation

While a larger dataset was collected, this study will only focus on the M-mode thoracic US captures [7]. For each subject, there were five M-mode images (1000 × 400 resolution) with a 5 s window of negative and injury-positive conditions. Injury-positive conditions were introduced following euthanasia by previously developed methods [7]. Briefly, to create positive PTX and HTX conditions, fluid or air was delivered with a catheter inserted between the pleural layers. Scans were collected with a standard Fujifilm Sonosite PX ultrasound, except for the last three subjects, whose scans were collected via video stream through a capture card. All US captures were recorded for approximately 30 s at 30 frames per second.

Dataset curation labels included animal subject ID, injury state, injury severity, signal quality index, and transducer steadiness. The injury severity for the HTX and PTX classes was graded by magnitude of injury as “positive” or “slight”. Signal quality index used a Likert scale to rank data based on whether the relevant anatomical features were in view, with 1 corresponding to poor quality and 5 representing best quality. If the US capture was noisy due to motion artifacts, this was noted. Subjective metrics such as steadiness and signal quality index were scored by two reviewers involved in US image capture. Only signal quality scores of 3 or above without motion artifacts were used for DL model development.

#### 3.1.2. Data Preprocessing

The entire dataset was split evenly across each animal protocol into three groups for the leave-one-split-out (LOSO) cross validation setup. The three groups were analyzed for image property distribution before training any DL model. A previous study [7] did not perform this analysis prior to implementing the LOSO cross validation methodology. 

Distribution analysis was performed to identify image-level and group-level differences, using Python libraries scikit-learn (ver. 1.5.1) and Matplotlib (ver. 3.8.3) on Python 3.11.7. The images were analyzed for standard image properties: brightness (B), or the average pixel intensity of the image, contrast (C), or the standard deviation of the pixel intensities of the image, and kurtosis (K), or the pixel intensity distribution of the image. Pixel intensity metrics were calculated and normalized using Z-standardization to plot and observe the relationship between the metrics of the images for the entire distribution of US captures. The evaluated metrics were plotted against each other on two-dimensional plots to observe differences between the images using contrast vs. brightness and brightness vs. kurtosis plots. An example of outlier US scans is shown in Figure 3.

Next, a confidence interval was defined on the US captures as a region within the Mahalanobis distribution of captures [25]. The squared Mahalanobis distances, or the distances between a US capture and its distribution, were calculated using the covariance of the metrics. Then, using the cumulative distribution function of the squared Mahalanobis distances, different confidences were evaluated to see US captures that were within the distribution. After choosing a confidence interval of 97.7%, labels for the dataset were generated depending on which data points fit within the interval, acting as a filter for the dataset. This step was performed on the two metric relationships: C vs. B and B vs. K.

### 3.2. DL Model Training and Evaluation

A training pipeline was created for US image label compilation that applied filter processing, configured the DL model architecture with data augmentations, and set up model training with different combinations of hyperparameters. DL model development was conducted with the PyTorch (ver. 2.2.0) framework using augmentation transformations to prevent overfitting, including a 50% probability that the image will be flipped, a brightness adjustment defined within 40% to 140% of the baseline value, and a contrast adjustment within 95% to 130% of the baseline contrast. The model architecture chosen for training was MobileNetV3 [26], due to previous success with utilizing MobileNetV2 [7,23] architecture to train binary classification models for injury interpretation For all training iterations, ImageNet1 kV1’s pre-trained weights were used as a starting point with the MobileNetV3 architecture. By default, the MobileNetV3 architecture defines its classifier with 1000 outputs or classes as per the ImageNet dataset. The fourth layer of the classifier that defines output size was replaced with a linear layer to define three outputs for the negative, HTX and PTX classes, thus changing the output of the dense layer. Training iterated over group splits for LOSO cross validation. DL models were trained on a 70:20:10 split of data for training, validation, and testing per LOSO fold. The accuracy of each LOSO fold was evaluated after the training and validation processes, using the sci-kit learn library to generate confusion matrices and accuracies for classifications.

#### 3.2.1. Hyperparameter Tuning

Based on previous studies, 100 training epochs with a batch size of 32 were used, as this is generally considered a good balance between variance in gradient estimates and convergence speed for computational efficiency and was used as a default value in the MobileNetV3 framework described and tested by Howard et al. [26,27]. A learning rate of 0.001 was chosen for training. Once training finished, the results and metadata regarding the LOSO folds were saved. This was repeated for all iterations of training for each preprocessing filter. In combination with filter group options, different hyperparameters were tested, such as weighted loss, validation patience, and weighted decay. Combinations of these hyperparameters were tested iteratively until a target testing accuracy of 85% across all LOSO folds was met. To account for class imbalances in the dataset, weighted loss was applied by calculating the frequency of classes per label and deriving a class weight inversely proportional to the frequency of the class using the sci-kit learn library. From this, the classes with lower frequency were given more weight during training to address the class representation imbalance of the dataset. Performance metrics were calculated with blind test data, with accuracy being calculated two ways—balanced and global accuracy. Balanced accuracy normalizes weights for each classification, while global accuracy does not account for class imbalances among HTX, PTX, and negative.

#### 3.2.2. Real-Time Testing

To further validate the performance and inference capabilities of the trained models, a Python script was developed to obtain US video captures, crop videos, and extract frames to inference the DL models while simulating real-time deployment on blind test images. Frames were extracted at 30 frames per second from the 10 s in the middle of each 30 s US video. The accuracy of each LOSO model was compared against prior YOLOv8 image classification-trained thoracic models, which were evaluated in real-time during large animal studies [8,28,29]. In addition, Gradient-weighted Class Activation Mapping (GradCAM) overlayed images were generated to show gradient hotspots for image regions of importance to AI predictions, to highlight model explainability [30].

## 4. Results

### 4.1. Cluster Analysis Results

The first analysis conducted evaluated the effects of preprocessing analysis for image filter development. The overall property distributions of the brightness (B), contrast (C), and kurtosis (K) metrics used for filter development are shown in Figure 4a. From the C vs. B relationship, it was evident that some US captures were separated as outliers from the rest of the population of the dataset (Figure 4b).

Highlighted in the C vs. B plot are two subjects with a cluster of US scans that are outliers to the standard image distribution. These images were substantially brighter/dimmer, and their variability could impact model training generalizability. Similarly, in Figure 4c, additional US captures based on swine subjects were identified as outliers from the standard population based on K vs. B relationship trends. Lastly, the K vs. C relationship trends had a similar distribution to the K vs. B plot, and therefore were not used for preprocessing methods in this study.

After testing different intervals for a confidence ellipse, it was found that a confidence interval of 97.7% excluded outliers while minimizing the loss of informative, representative captures. Next, Figure 5a,b illustrates how the selected confidence interval established the general shape of the confidence ellipse and which data points were filtered out of the ellipse based on their location on the plot. Figure 5a shows how images with brightness and contrast outside of the 97.7% confidence interval (approximately 2 arbitrary units [a.u.] from the centroid) were excluded. Many images with low contrast and brightness were still within the confidence ellipse, and were thus included via this C vs. B filter. From Figure 5a, images that were significantly high either in brightness (2.5 a.u.) or in contrast (2.8 a.u.) were not considered fit for the K vs. B ellipse based on the 97.7% confidence intervals defining the ellipse shape.

### 4.2. AI Model Development

AI models were developed using LOSO cross validation with a range of hyperparameter settings to identify which model setup performed best with real-time (RT) datasets. A summary of model performance for global and balanced accuracy across all hyperparameter configurations and filters is shown in Table 1. The best-performing models using a C vs. B filter scored a global accuracy of 84.74%, with an average balanced accuracy at 90.49%. In comparison, the best-performing models that used no filter scored average global and balanced accuracies of 86.63%, and 88.17%, respectively. Conversely, the least accurate model from the average scored a balanced accuracy of just 78.63%.

Next, balanced accuracy model performance was compared within each LOSO fold, per preprocessing filter used (Figure 6). “No Filter” models achieved high accuracies above 85% in several LOSO models, with LOSO fold 3 achieving 95% balanced accuracy, most notably for one model configuration. With the C vs. B filter, the highest-performing model achieved over 95% balanced accuracy, and the average performance for LOSO folds 1 and 3 were higher than the “No Filter” group. However, C vs. B also had the poorest scoring model, from LOSO fold 2, at 59% balanced accuracy.

From the real-time performance results illustrated in Table 1 and Figure 6, the LOSO models with the best average balanced accuracy were selected. In this case, models that used the C vs. B filter, 10 epoch validation patience, balanced weighted loss, and a 0.00001 weighted decay value achieved the best average balanced accuracy, of 90%. Figure 7a reflects the evaluation of the best performing model on 8677 negative images, 1498 HTX images, and 2993 PTX images, averaged across three LOSO folds. The classification imbalance was accounted for by including the weighted loss hyperparameter mentioned in Section 3.2.1, in order to give more weight to the captures with less frequency in the dataset. For the negative class, the models correctly identified negative classes for 6790 captures; however, they incorrectly predicted HTX or PTX for the other 1887 negative captures, resulting in a recall of 78% (Figure 7c). The models incorrectly predicted false positive results for 26 captures for the negative class, scoring a precision of 99%. In the HTX class, true positive predictions were made for 1418 captures, achieving a recall of 95% and a low precision of 54%. The PTX class yielded a recall score of 99% and a precision of 79%, as most of the PTX captures were correctly identified, and some of the PTX predictions were made on the negative captures.

Model predictions were assessed using GradCAM overlays for each correct class prediction (Figure 7b). For the negative class, the model is evaluating predictions by observing the lung motion due to breathing under the pleural line. The HTX true positive GradCAM overlay observes the dark space above the pleural line, tracking along the area where fluid would accumulate physiologically. Lastly, the PTX true positive GradCAM overlay closely tracks the pleural line, with some bias towards the top of the image due to tracking the x-axis tick marks.

Next, the differences between training and real-time testing accuracy for the best-performing model from this study were compared to models developed in a previous study (Figure 8) [8]. There was a large discrepancy between the previous study’s training and real-time performance, with the real-time being 19% worse than the training accuracy, at 86%. For this study, the training accuracy for global accuracy scored lower, at 83%, but the real-time scoring was consistent at 85%, highlighting the model’s improved generalization for this application. Between training and real-time streamed M-mode results, there was a notable difference of 6.82% between average balanced accuracy in training and real-time evaluations, whereas the difference in global accuracy was less than 2%.

## 5. Discussion

Image interpretation from US medical imaging is critical for triaging injuries and guiding diagnosis during emergencies and in prehospital settings. AI can automate these interpretation decisions if proven effective on blind-tested datasets in real-time implementations. This study explored the effects of different preprocessing methods and DL parameter functions for developing a three-class thoracic DL classification model.

In previous research efforts for DL model development [8], there was a large disparity between real-time and training accuracy, highlighting the need to develop more generalized DL models. To address this, the current study focused on improved image analysis preprocessing techniques and incorporated model training parameters to counteract some of the limitations of the dataset, which the last study did not explore for thoracic scan sites. Real-time performance increased to 86%, compared to only 67% in the previous real-time study, by implementing the above improvement. Training performance was comparable between both studies, but blind test performance was significantly improved in this study, suggesting that the filters and hyperparameters used helped the models with better generalization for this thoracic image classification application.

Of note, there were differences between failure cases for MobileNetV3 during training and real-time use. During training (Figure 8b), MobileNetV3 had strong recall scores for each of the three prediction classes, each above 80%. The lowest precision metric was PTX, with 14 false positives compared to 54 true positive predictions. In real time, recall remained high, with HTX and PTX having stronger scores, while the score for the negative class slightly dropped from 81% to 78%. False class identification for negative images was skewed towards HTX over PTX. Larger changes were evident with precision metrics. Negative precision was significantly improved, and PTX remained the same. However, HTX dropped from 82% to 54%, primarily due to false positive HTX predictions from true negative class images. It is unclear why this bias was more pronounced during real-time image capture, but future model development should be focused on overcoming this model shortcoming.

Without the use of a preprocessing filter, a weighted loss function, or regularization, the LOSO folds performed with an average global accuracy of 88%, although with lower precision and recall, indicating bias towards predicting the negative class. There were fewer data for HTX and PTX compared to negative US captures; therefore, global accuracy is skewed toward the negative class. After using the preprocessing C vs. B filter and tuned DL model hyperparameter functions, significant improvements were found in average recall or “balanced” accuracies for each model. This indicates that having an applied confidence region for US scan metrics helped the model train and learn appropriate features representing the population. Weighted loss was integrated in model training as a function of loss to mitigate the effects of class imbalance and help prevent overfitting, which was apparent when weighted loss was not used. Overall, the C vs. B filter was more effective than the K vs. B filter. This may be due to kurtosis as a textural metric not correlating as well as brightness or contrast—intensity metrics—when it comes to the features in the M-mode thoracic US and how their intensity values change when there is air or blood present in the pleural space for the injury-positive classes.

While some hyperparameter tuning and filter preprocessing techniques were tested, there are other potential techniques to explore that may help the models achieve higher performance. For example, for the context of this study and the importance of inference speed of the models, MobileNetV3 was chosen, since it is lightweight, and thus significantly faster at making predictions than other comparable architectures. With the methods and preprocessing techniques developed from this study, other models can be explored or developed from scratch through Bayesian optimization strategies [31,32]. Next, fixed weights for the weighted loss of each class can be explored to evaluate for any model performance improvements, instead of using a function that calculates this based on input frequency. In addition, the only metrics used for creating the filters were brightness, contrast, and kurtosis; however, other kinds of image-based metrics can be prepared to help identify other symptoms of a shift in data distribution. While each of these approaches may improve training accuracy, the developed models surpassed the target accuracy for this application of 85% and were thus not explored further. This blind accuracy is comparable to other studies for detection of lung motion from M-mode images that have achieved 82.4% in real-time clinical images [33] or 89% accuracy when paired with segmentation models [34]. The uniqueness of the models trained in this study is that they can differentiate between three injury states, classifying US M-mode images in three classes—HTX, PTX, and injury-negative. 

There are limitations to be considered regarding the scope of this study. First, there are data limitations to the M-mode thoracic dataset. Overall, the total number of M-mode US captures is generally much lower than B-mode image captures, primarily due to each M-mode capture taking 5 s to acquire a single image, while a video clip of the same length contains possibly hundreds of frames. The limitation can be further distilled to just the M-Mode US captures, as there were significantly fewer captures for the HTX and PTX classifications. This poses some questions on how well the population of these injuries is represented by the dataset. Further fine-tuning and other preprocessing methods can help with addressing how well each model predicts the underrepresented classes. Another limitation of this study is that real-time data predictions were performed retrospectively instead of in true real-time fashion. This is a limitation of animal studies having set time points that did not align with DL model development, but the streamed US image capture approach is analogous to how these approaches will be initially deployed for use in real-time applications.

## 6. Conclusions

Image classification DL models can improve the utility of ultrasound implementation for improving medical imaging-based triage for emergency medicine and prehospital applications. Implementation, however, requires DL models suitable for removing outlier data and generalized to a wide variety of use cases for real-time application. By evaluating the characteristics of thoracic M-mode US captures, this effort determined that there was value in implementing additional preprocessing techniques and model parameter functions to reduce variability in performance. This resulted in maintaining an average balanced accuracy of 90.49% and an average global accuracy of 84.74%, which demonstrates the models’ utility to resolve some of the challenges with making thoracic injury diagnostics in a pre-hospital setting. With these models running on a MobileNetV3 architecture, streamlining a diagnostic procedure with fast inferencing time in a pre-hospital setting can be especially useful for improving triage efforts.

## Figures and Tables

**Figure 1 jimaging-11-00222-f001:**
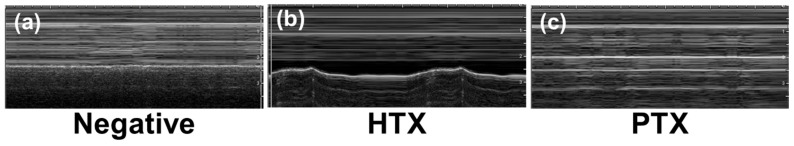
Representative motion mode (M-mode) US images of relevant thoracic region injury states. Images shown are for (**a**) baseline or negative for injury, (**b**) hemothorax or HTX, and (**c**) pneumothorax or PTX.

**Figure 2 jimaging-11-00222-f002:**
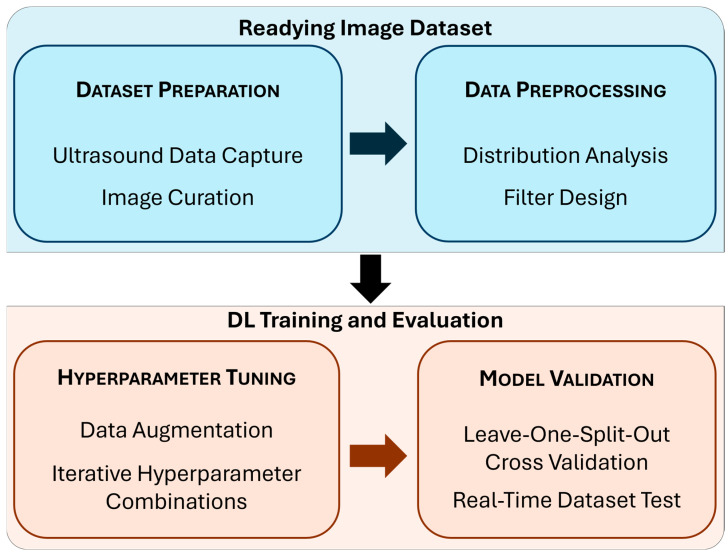
Block diagram summarizing the study methods. Each subsection below includes a detailed description of the phases of the study highlighted in this diagram.

**Figure 3 jimaging-11-00222-f003:**
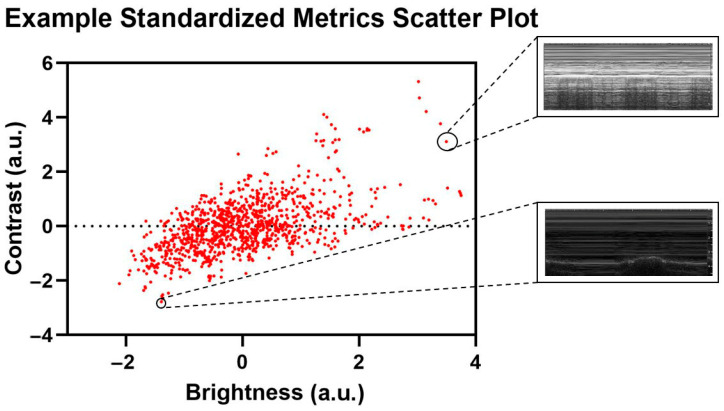
Representative standardized metric scatter plot with corresponding US captures. Selected US captures are shown from the corresponding data points in the scatter plot.

**Figure 4 jimaging-11-00222-f004:**
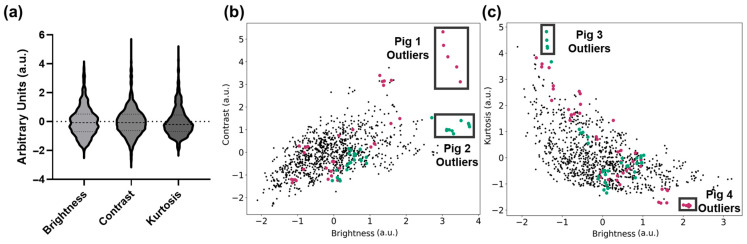
Standardized metric plots with pig labels. (**a**) Violin plots highlighting the data distribution for brightness, contrast, and kurtosis. Dashed lines for the median and first and third quartile for each metric are shown. (**b**) Contrast and brightness plot highlighting US captures that belong to Pig 1 (dark pink) and Pig 2 (teal), with outlier groups (black boxes). (**c**) Kurtosis and brightness plot highlighting groups of US captures that belong to Pig 3 (teal) and Pig 4 (dark pink), with outlier groups (black boxes).

**Figure 5 jimaging-11-00222-f005:**
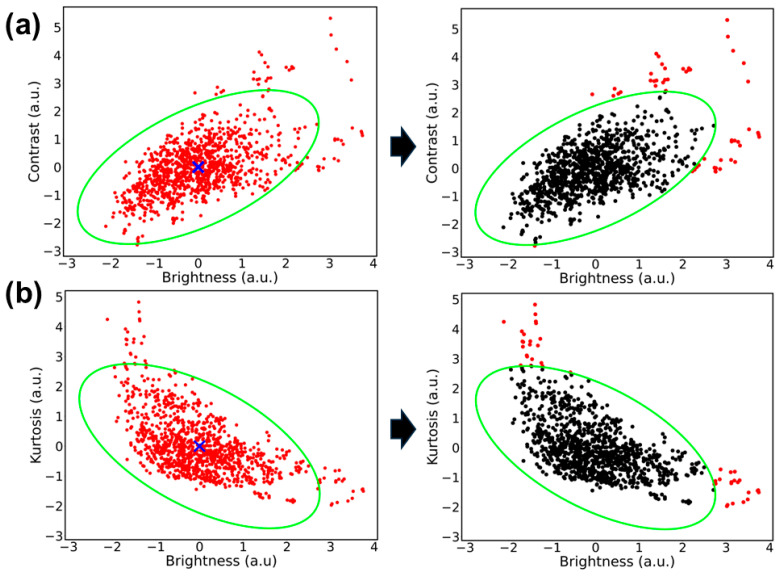
Plot of image metrics with corresponding confidence ellipse filter. (**a**) Contrast vs. brightness metrics and (**b**) kurtosis vs. brightness metrics. (Left image). Standardized metrics plotted for each thoracic M-mode US (red dots) with the calculated centroid (blue “x”) and confidence ellipse overlayed (bright green). (Right image) Differentiates the US captures that are within the confidence ellipse (black) and which corresponding US captures are excluded from the filter (red).

**Figure 6 jimaging-11-00222-f006:**
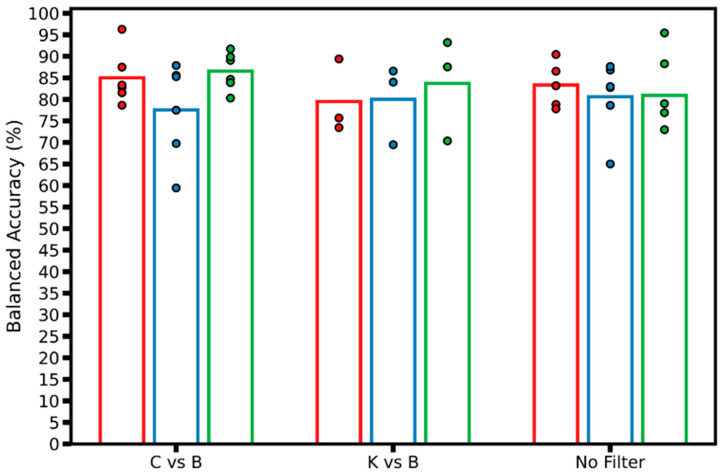
Balanced accuracy model performance plot. Real-time performance of models by group of filters by LOSO; each data point represents a different model trained, with the corresponding LOSO fold (N = 3) shown as different colors. The hollow bar plot represents the average of all models trained for each LOSO across each group of filters.

**Figure 7 jimaging-11-00222-f007:**
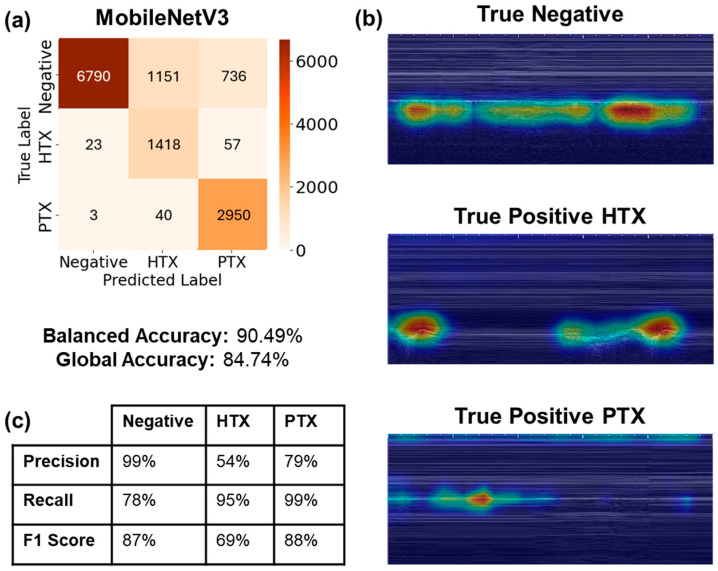
Real-time performance visualization metrics. (**a**) Three-class confusion matrix of the three LOSO models evaluated for the model with best balanced accuracy. Confusion matrix for blind-streamed results predicted US captures exported from 44 streamed videos. About 300 frames were extracted per video to represent 10 s of capture. (**b**) Representative GradCAM visualizations from LOSO fold 1 predictions for true negative, true positive HTX, and true positive PTX cases. (**c**) Summary table of precision, recall, and F1 score metrics for each classification label.

**Figure 8 jimaging-11-00222-f008:**
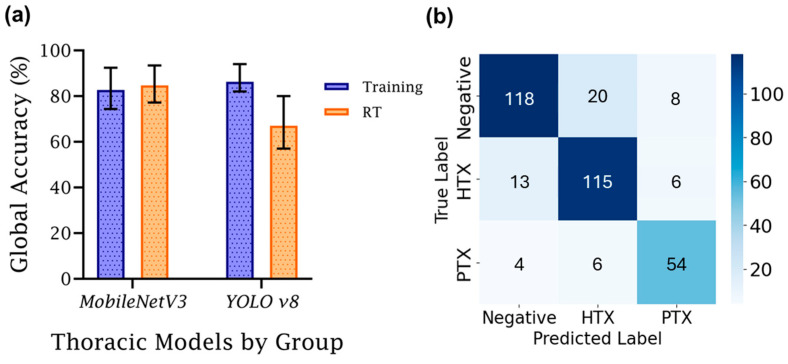
Comparison of training and real-time accuracy for M-mode DL classification models. (**a**) Training vs. real-time (RT) global accuracy, grouped by model architecture used in this study (MobileNetV3) and a former study (YOLOv8). Average training performance is denoted with an error bar for each group denoting standard deviation (*n* = 3). (**b**) MobileNetV3 training averaged LOSO confusion matrix.

**Table 1 jimaging-11-00222-t001:** Summary of the model accuracy for each hyperparameter configuration. Results are shown for balanced and global accuracy as average and standard deviation across each LOSO split. Bolded rows indicate the highest performing hyperparameter configurations.

Filter	Weighted Loss	Validation Patience	Weighted Decay	Balanced Accuracy AVG (%)	Balanced Accuracy STD (%)	Global Accuracy AVG (%)	Global Accuracy STD (%)
**No Filter**	**None**	**None**	**None**	**88.17**	**7.05**	**86.63**	**7.97**
No Filter	None	10	None	84.64	4.14	84.64	5.08
No Filter	Balanced	None	None	76.86	8.91	76.86	10.91
No Filter	Balanced	10	None	79.64	4.72	79.64	5.79
No Filter	Balanced	10	1 × 10^−4^	80.8	4.85	80.67	6.04
No Filter	Balanced	10	1 × 10^−5^	79.7	4.76	79.7	5.83
C vs. B	None	None	None	82.89	2.16	87.8	0.95
C vs. B	None	10	None	79.55	14.32	81.16	7.6
C vs. B	Balanced	None	None	80.27	3.15	75.34	5.85
C vs. B	Balanced	10	None	86.75	2.47	84.96	11.29
C vs. B	Balanced	10	1 × 10^−4^	78.39	6.18	78.75	12.91
**C vs. B**	**Balanced**	**10**	**1 × 10^−5^**	**90.49**	**4.51**	**84.74**	**8.18**
**K vs. B**	**None**	**None**	**None**	**85.17**	**7.23**	**84.87**	**3.05**
K vs. B	Balanced	10	None	81.26	8	81.26	9.8
K vs. B	Balanced	10	1 × 10^−5^	76.83	7.75	81.15	12.31

## Data Availability

The data presented in this study are not publicly available because they have been collected and maintained in a government-controlled database located at the U.S. Army Institute of Surgical Research. This data can be made available through the development of a Cooperative Research and Development Agreement (CRADA) with the corresponding author.

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
