# Peer review of "Development of Deep Learning Models for Real-Time Thoracic Ultrasound Image Interpretation"

_2313-433X, 2025, doi:10.3390/jimaging11070222_

Round 1

Reviewer 1 Report

Comments and Suggestions for Authors
  1. Focus on Thoracic Injury Detection
  2. Cluster analysis was used to define the “population” based on brightness, contrast, and kurtosis properties. - As mentioned in the paper the normalization was necessary. 
  3. Properly associate the “m-mode” with its meaning. Readers of the paper might not understand what m-mode refers to. The long version is mentioned in the Abstract, but link the acronym somewhere in the Introduction.
  4. Line 112: The dataset was split into three groups of subjects, split evenly across each animal pro-112 tocol. This sentence is not clear: what are the three groups?
  5. Line 183: May consider to rewrite that line.
  6. Provide the acronym HTX of hemothorax at least once in the text, including Introduction etc. Mentioning only once and only in the Abstract is not good.
  7. Strongly suggest to show representative images for the three classes in the Introduction or Methods chapter. Readers need to see the type of images, the texture, brightness

Comments on the Quality of English Language

Some odd choice of word is observed. 

Using some heavy wording, "Readying the dataset" is not very appropriate. 

Image "captured", could be used as image "acquired". 

Author Response

  1. Focus on Thoracic Injury Detection

Thanks for taking the time to review our manuscript. Yes, that is correct, this study is focused only on thoracic injuries that would present during trauma cases.

  1. Cluster analysis was used to define the “population” based on brightness, contrast, and kurtosis properties. - As mentioned in the paper the normalization was necessary. 

Yes,  that is correct again – normalization of the dataset allowed for proper comparisons of the populations which were used to improve the model performance.

  1. Properly associate the “m-mode” with its meaning. Readers of the paper might not understand what m-mode refers to. The long version is mentioned in the Abstract, but link the acronym somewhere in the Introduction.

Good suggestion, we updated the introduction to reflect the long version of the M-mode name.

  1. Line 112: The dataset was split into three groups of subjects, split evenly across each animal protocol. This sentence is not clear: what are the three groups?

We have added more clarity to better explain what the groups are for this experimental setup.  

  1. Line 183: May consider to rewrite that line.

I believe this was part of the template from the journal and has been removed as it was left in by accident.

  1. Provide the acronym HTX of hemothorax at least once in the text, including Introduction etc. Mentioning only once and only in the Abstract is not good.

Yes, we have now defined the HTX algorithm also in the introduction.

  1. Strongly suggest to show representative images for the three classes in the Introduction or Methods chapter. Readers need to see the type of images, the texture, brightness

Good suggestion – we have added example images for all three classes to an introductory figure.

Reviewer 2 Report

Comments and Suggestions for Authors

In this work titled “Development of Deep Learning Algorithm Models for Real-Time Thoracic Injury Detection”, a MobileNet deep learning model has been trained to classify ultrasound images of Thoracic injury into three classes of pneumothorax (PTX), hemothorax (HTX) and Negative of Swine (or Pig) subjects. The model has been trained across various hyperparameters (weighted loss, validation patience, and weighted decay) and filters (Contrast vs Brightness; Kurtosis vs Brightness). Leave-one-split-out (LOSO) cross-validation has been used to assess the performance of the model. For the test data with filter of Contrast vs Brightness and with hyperparameters values of Balanced, 10 and 1.00E-05 for weighted loss, validation patience, and weighted decay showed highest Balanced accuracy of 90.49% and Global accuracy of 84.74%.

But, address the following questions/suggestions.

  1. It is limited to using pre-trained Deep Learning (DL) architecture, MobileNet, for classification of Point-of-Care-Ultrasound (POCUS) images. But, why not the other light weight models such as SqueezeNet and ShuffleNet were not considered for the comparison study.
  2. Under the “Abstract” section, line number 12, provide the expansion of POCUS after the word ultrasound i.e., Point-of-Care-Ultrasound (POCUS). Expansion at once is enough, hence no requirement at line number 34. MobileNet architecture is a Deep Learning (DL) model. Hence refer to it as DL model instead of AI model (line number 22). Similar through out the paper it is a suggestion to referring the MobileNet as DL model.
  3. Title of paper needs to be modified because it is not on Injury detection (i.e. it is not a study on detection where the injury is within the image). The work discussed in this paper is on Classification. It is suitable to change title appropriate to work. In the title “Real-Time” has been used, but this needs to be clarified because the trained model has been tested on the testing data. Or justification can be provided what the term “Real-Time” means specific to this paper work.
  4. At the end of “Introduction” section i. Objectives of the work should mentioned and ii. The structure of the paper can be provided.
  5. There is no separate section on “Literature Survey”, Why ?.  Authors should discuss on the results obtained by already published works in the similar area.
  6. Write up of the paper needs to improved. i. Avoid the words such as Our, We (Line number, 57, 61). ii. Line number 321, Overall, The C vs B filter….  iii. Use the brackets () instead of [] for arbitrary units [a.u.]
  7. Figure 1, Provide further description on the terms “Metric Scoring”, “Distributed Analysis”, “Filter Design” and “Filter Type”
  8. Under the Section 3.2, line number 253, “AI Model Development”, 8677 Negative images, 1498 HTX 253 images, and 2993 PTX images has been used for the study. There is an imbalance in data among the three classes. Hence, discussion on how to handle imbalance of data is required.
  9. Section 2.1.1, “The dataset was split into three groups of subjects”, What are three groups of subjects?
  10. Figure 2, instead of giving axis labels as Metric 1 and Metric 2, it is preferred to give the appropriate name of the metric as axis label
  11. Section 2.2, “AI Model Training and Evaluation”, the pre-trained MobileNet model on ImageNet data has been used. But, authors should provide details on which existing layers of the pre-trained MobileNet model have been removed and which layers were finetuned. And how the last Dense layer has been modified.
  12. Section 2.2, line number 171 the class labels mentioned are HTX, PTX and Negative. But, this conflicts with class labels mentioned as “Positive”, “Slight” or  “Negative” for injury severity in line number 105.  Provide clarification in this regard.
  13. For figures 3, 4 and 6, provide sub figure numbers as (a), (b), (c)… as figure caption at the bottom of the figure. Instead of the subfigure numbers as A, B, C at the top of the figure.
  14. Figure 7, why the comparison of results of MobileNet model with YOLO model. Because MobileNet is an image classification deep learning model but YOLO is object detection deep learning model.
  15. Under “Reference” listing, preferred not to use the following self-cited papers.

[11] S. I. Hernandez-Torres, C. Bedolla, D. Berard, and E. J. Snider, “An extended focused assessment with sonography in trauma ultrasound tissue-mimicking phantom for developing automated diagnostic technologies,” Frontiers in Bioengineering and Bio-technology, vol. 11, p. 1244616, 2023. 413

[12] E. N. Boice, S. I. Hernandez-Torres, Z. J. Knowlton, D. Berard, J. M. Gonzalez, and E. J. Snider, “Training Ultrasound Image Classification Deep-Learning Algorithms for Pneumothorax Detection using a Synthetic Tissue Phantom,” Journal of Imaging, vol.8, no. 249, 2022.

[13] S. I. Hernandez Torres, A. Ruiz, L. Holland, R. Ortiz, and E. J. Snider, “Evaluation of Deep Learning Model Architectures for Point-of-Care Ultrasound Diagnostics,” Bioengineering, vol. 11, no. 4, Art. no. 4, Apr. 2024, doi: 10.3390/bioengineering11040392.

[14] S. I. Hernandez Torres et al., “Real-Time Deployment of Ultrasound Image Interpretation AI Models for Emergency Medi-cine Triage Using a Swine Model,” Technologies, vol. 13, no. 1, Art. no. 1, Jan. 2025, doi: 10.3390/technologies13010029.

Author Response

But, address the following questions/suggestions.

  1. It is limited to using pre-trained Deep Learning (DL) architecture, MobileNet, for classification of Point-of-Care-Ultrasound (POCUS) images. But, why not the other light weight models such as SqueezeNet and ShuffleNet were not considered for the comparison study.

We appreciate the reviewer taking the time to review our manuscript. MobileNet was the focus as we have previously completed a study comparing both large and lightweight models and Mobilenet outperformed many others. The only difference was MobileNetV3 was used here vs. prior MobileNetV2. This is why the others are not compared in this study. This is mentioned in Section 2.2 in the methods.  

  1. Under the “Abstract” section, line number 12, provide the expansion of POCUS after the word ultrasound i.e., Point-of-Care-Ultrasound (POCUS). Expansion at once is enough, hence no requirement at line number 34. MobileNet architecture is a Deep Learning (DL) model. Hence refer to it as DL model instead of AI model (line number 22). Similar through out the paper it is a suggestion to referring the MobileNet as DL model.

We have fixed the POCUS acronym first mention as the review recommended. We do still re-define the POCUS acronym at first mention in the introduction based on other reviewer feedback with regards to how acronyms should be handled. We defer to the editorial staff for the journal if our manuscript becomes suitable for publication. We also took the reviewer’s suggestion and modified AI to DL where appropriate throughout the manuscript.

  1. Title of paper needs to be modified because it is not on Injury detection (i.e. it is not a study on detection where the injury is within the image). The work discussed in this paper is on Classification. It is suitable to change title appropriate to work. In the title “Real-Time” has been used, but this needs to be clarified because the trained model has been tested on the testing data. Or justification can be provided what the term “Real-Time” means specific to this paper work.

We modified the title to “Development of Deep Learning Models for Re-al-Time Thoracic Ultrasound Image Interpretation” to hopefully address some of the confusion. Real-time captured data is still the used and the focus of testing in this manuscript so we did not modify that verbiage in the title, however.

  1. At the end of “Introduction” section i. Objectives of the work should mentioned and ii. The structure of the paper can be provided.

We summarized the objective and main approaches taken in the final paragraph on the introduction. We hope that is suitable for addressing this issue. The structure of the paper is pretty standard – introduction, methods, results, discussion, and conclusions – so we did not add this summary to the introduction.

  1. There is no separate section on “Literature Survey”, Why ?.  Authors should discuss on the results obtained by already published works in the similar area.

Thanks for the suggestion, we have added more details in the introduction comparing the planned work for this study to other published works. We hope this is suitable to address this concern.

  1. Write up of the paper needs to improved. i. Avoid the words such as Our, We (Line number, 57, 61). ii. Line number 321, Overall, The C vs B filter….  iii. Use the brackets () instead of [] for arbitrary units [a.u.]

Thank you for the recommendations, we have thoroughly reviewed the manuscript and removed uses of first person, and corrected capitalizations where appropriate. The use of [ ] was a result of defining that acronym within a ( ) which we believe is the appropriate convention. However, we defer to the editorial staff on their preference if our manuscript is suitable for publication.

  1. Figure 1, Provide further description on the terms “Metric Scoring”, “Distributed Analysis”, “Filter Design” and “Filter Type”

As figure 1 is meant to be an overview, summarizing the order of events in the study, the terms mentioned are further explained in their respective subsections. We added this statement to the pre-amble of the methods section as well.

  1. Under the Section 3.2, line number 253, “AI Model Development”, 8677 Negative images, 1498 HTX 253 images, and 2993 PTX images has been used for the study. There is an imbalance in data among the three classes. Hence, discussion on how to handle imbalance of data is required.

We acknowledge the class imbalance issue with our dataset. This was accounted for during training through weighted loss functions. This is described in 2.2.1, and we have added further clarifiers to make sure the two points are connected. The justification for the this approach addressing class imbalance is also described in the Discussion section.  

  1. Section 2.1.1, “The dataset was split into three groups of subjects”, What are three groups of subjects?

This section has been re-worded for clarity to better explain what these groups represent. They are crossfold validation splits used throughout training.

  1. Figure 2, instead of giving axis labels as Metric 1 and Metric 2, it is preferred to give the appropriate name of the metric as axis label

This figure has been modified to describe what two metrics are used per reviewer recommendation.

  1. Section 2.2, “AI Model Training and Evaluation”, the pre-trained MobileNet model on ImageNet data has been used. But, authors should provide details on which existing layers of the pre-trained MobileNet model have been removed and which layers were finetuned. And how the last Dense layer has been modified.

Thanks for the suggestion, the final layer of the model was modified to allow for the smaller number of classes used in this study vs. the ImageNet pre-trained model. This has now been described in more detail in Section 2.2.

  1. Section 2.2, line number 171 the class labels mentioned are HTX, PTX and Negative. But, this conflicts with class labels mentioned as “Positive”, “Slight” or  “Negative” for injury severity in line number 105.  Provide clarification in this regard.

There is a misunderstanding here – the classes for the AI model were HTX PTX or Negative only. The Positive Slight or Negative identifiers were during image curation to identify which images had small or insignificant injuries as these were not used during training datasets. We have updated the wording where the injury severities are mentioned for improved clarity.

  1. For figures 3, 4 and 6, provide sub figure numbers as (a), (b), (c)… as figure caption at the bottom of the figure. Instead of the subfigure numbers as A, B, C at the top of the figure.

We have modified the figure sub-parts to reflect the reviewer’s preference.

  1. Figure 7, why the comparison of results of MobileNet model with YOLO model. Because MobileNet is an image classification deep learning model but YOLO is object detection deep learning model.

YOLO is traditionally an object detection model but can be configured for image classification. That is how the model is formatted for this use case and was clarified in the Methods Section 2.2.2.

  1. Under “Reference” listing, preferred not to use the following self-cited papers.

As this work builds upon prior research efforts these were added to ensure the reader could better track the connection to prior work. However, we have minimized the quantity of these to only the most relevant to address this point.

Reviewer 3 Report

Comments and Suggestions for Authors

This research presents an investigation of deep learning algorithm frameworks for real-time thoracic injury diagnosis using an image classification approach. This technique utilized the MobileNetV3 artificial intelligence model framework to optimize several hyperparameters, and real-time image collection assessed the results. The presentation of the paper is well structured and covers almost each and every piece of required information to accept this paper for publication. But the research contribution is missing in the introduction section. We suggest inserting a comparison of your research contributions with existing state-of-the-art approaches at the end of the introduction section. After careful consideration of this suggestion, it is recommended to accept this paper for publication.

Author Response

We suggest inserting a comparison of your research contributions with existing state-of-the-art approaches at the end of the introduction section. After careful consideration of this suggestion, it is recommended to accept this paper for publication.

We appreciate the reviewer taking the time to review our manuscript and for the recommendation to accept it for publication. We have added more reference to state-of-the-art in a more detailed literature review in the introduction, as well as more explicitly mentioning the objective and research contributions for this study in the final paragraph of the introduction.